# Correlation cluster analysis of slope safety monitoring data in reservoir areas

**Weixing Yang**[1,2,3]*, **Tingting Li**[4], **Bo Wen**[1], **Yuan Ren**[1]

1 Power China Northwest Engineering Corporation Limited, Xi'an, China, 2 Experimental Research Department, Technical Centre, Northwest Survey and Design Research Institute of China Electric Construction Group Co, 3 Safety Intelligent Monitoring Innovation Workshop, 4 Xi'an Children's Hospital, Xi'an, China

\* 75604366@qq.com

## Abstract

Current predictive methods for dam failures in reservoirs remain limited, indicating that the underlying mechanisms of such failures are not yet fully understood. To further elucidate the interrelationships among safety monitoring data in the reservoir area, this study established 36 monitoring cross-sections distributed across upper, middle, and lower slope zones. Each cross-section was instrumented with eight different types of monitoring devices. A total of 4,320 samples were collected (432 samples per instrument type), resulting in an overall dataset of 34,560 measurements. The monitoring data were sequentially analyzed using: (1) descriptive statistics, (2) Welch/Brown-Forsythe post hoc One-way analysis of variance (ANOVA), and (3) cluster analysis. The results demonstrate that: (1) Significant correlations exist among monitoring variables, with the strongest positive correlation observed between loading and lean ($r = 0.40$), while the strongest negative correlation occurred between sedimentation and stress ($r = -0.39$). (2) Cluster analysis of the eight monitoring variables revealed two distinct clusters: soil displacement, stress, and water-level formed one cluster, while the remaining variables comprised the second cluster. In summary, variations in monitoring data and their correlations resulted from water-level and environmental changes in the reservoir area, with spatial differences across monitoring types. A thorough investigation of these variations and their causes will enable accurate safety assessments of the reservoir area and support tailored response strategies for different locations.

## 1 Introduction

Reservoir collapse is the leading cause of loss of life and property of people downstream of reservoirs [1–5]. Since 2024, global climate change has contributed to reservoir dam failures, significantly impacting local populations' lives and property as major safety incidents [3,6–10]. Despite implementing at least three preventive

**Data availability statement:** All relevant data are within the manuscript and its Supporting Information files.

**Funding:** The author(s) received no specific funding for this work.

**Competing interests:** The authors have declared that no competing interests exist.

measures - slope reinforcement [1,2,11], load reduction at slope crests [12–15], and water-level control [16–18] reservoir dam failures remain unpreventable [8,19].

Current methods cannot reliably predict reservoir dam failures, suggesting their underlying causes remain unknown. While water-level fluctuations clearly impact reservoir-side communities [8,10], their effects on slope stability remain uncertain. Prolonged neglect of this situation may trigger safety hazards including landslides [5,10,20], soil displacement [17,21], and even reservoir dam failure [18,22]. Research indicates that water table fluctuations disturb subsurface soil stresses, inducing displacement that increases seepage and potentially leads to dam failure [11,23–25]. However, the potential relationships between soil disturbance, soil displacement, seepage, and dam failure remain unproven.

In order to further reveal the correlation between internal and external factors affecting slope safety in the reservoir area, we divided the slopes into 36 sections, with each section having three places classified as top, middle and bottom. Eight monitoring types are deployed at each site and observed over time. The correlation between monitoring types was further explored by statistically analyzing the data, and the correlation between monitoring types at different places was also investigated. Cluster analysis of different monitoring types was performed by normalizing different types of monitoring data, at the same time, sensitivity analyses are conducted for different monitoring types.

Our study revealed that seepage has lower sensitivity to reservoir safety than other factors, while soil displacement proved critical [26,27]. Stress was significantly negatively correlated with sedimentation and seepage, loading was also significantly negatively correlated with soil displacement, while loading was significantly positively correlated with tilt. The types of monitoring are divided into 2 main categories and those located in different places are divided into 4 main categories. These studies provide some references to further reveal the interactions and aggregation between the internal and external changes of the slope in the reservoir area.

## 2 Materials and methods

### 2.1 Study area

The study was conducted at the upper reservoir of a pumped storage power station in Xinzhou City, Shanxi Province, China. Located in the Shanxi Plateau within the central Taihang Mountain Range (northern terminus of the Zhoushan Mountain System), the study area features summit elevations of 1500–2500 m with 500–1000 m relief, characterizing alpine topography. The region experiences temperature extremes from -34°C to 36°C and receives 460 mm average annual precipitation.

The reservoir is situated in a canyon terrain characterized by steep slopes with elevation differences ranging from 500 m to 1000 m. The lithology predominantly consists of limestone, dolomite, and sandstone, with bedrock strata exhibiting a northeast-northeasterly strike and dipping outward from the reservoir area at angles of 6–10°. The reservoir infrastructure comprises a main dam, auxiliary dam, wave protection wall, water intake structure, and outlet structure. Operational parameters include a normal pool level of 1492.5 m above sea level and a dead water-level of

1467 m, yielding a total storage capacity of 4.851 million cubic meters. This includes a dead storage volume of 0.610 million cubic meters and a regulated storage capacity of 4.241 million cubic meters. Photographic documentation of the reservoir area and routine monitoring installations is provided in Fig 1.

### 2.2 Experimental design

**2.2.1 Distribution of monitoring points.** In order to facilitate the study of the interrelationships between the various areas, the reservoir area was systematically divided into 6 ordered zones (Area I-VI), each containing six monitoring sections stratified into top, middle, and bottom elevations for safety assessment. Eight monitoring types were deployed at each location: sedimentation, fissure, tilt, seepage, soil displacement, stress, loading, and water-level. The spatial distribution of monitoring points is illustrated in Fig 2 and Table 1.

**2.2.2 Data classification and preliminary analyses.** In order to analyze the data more rationally, the sample data would be classified in this study. Through preliminary analyses of the monitoring data and the distribution of monitoring points in the reservoir area over the past 10 years, we had broadly classified the monitoring data into two main categories: external monitoring and internal monitoring. External monitoring data were sedimentation, fissure and tilt; Internal monitoring data were seepage, soil displacement, stress, loading and water-level. The design of this study was to repeat the measurements four times for each monitoring type, with 96 sets of measurements collected for each section, for a

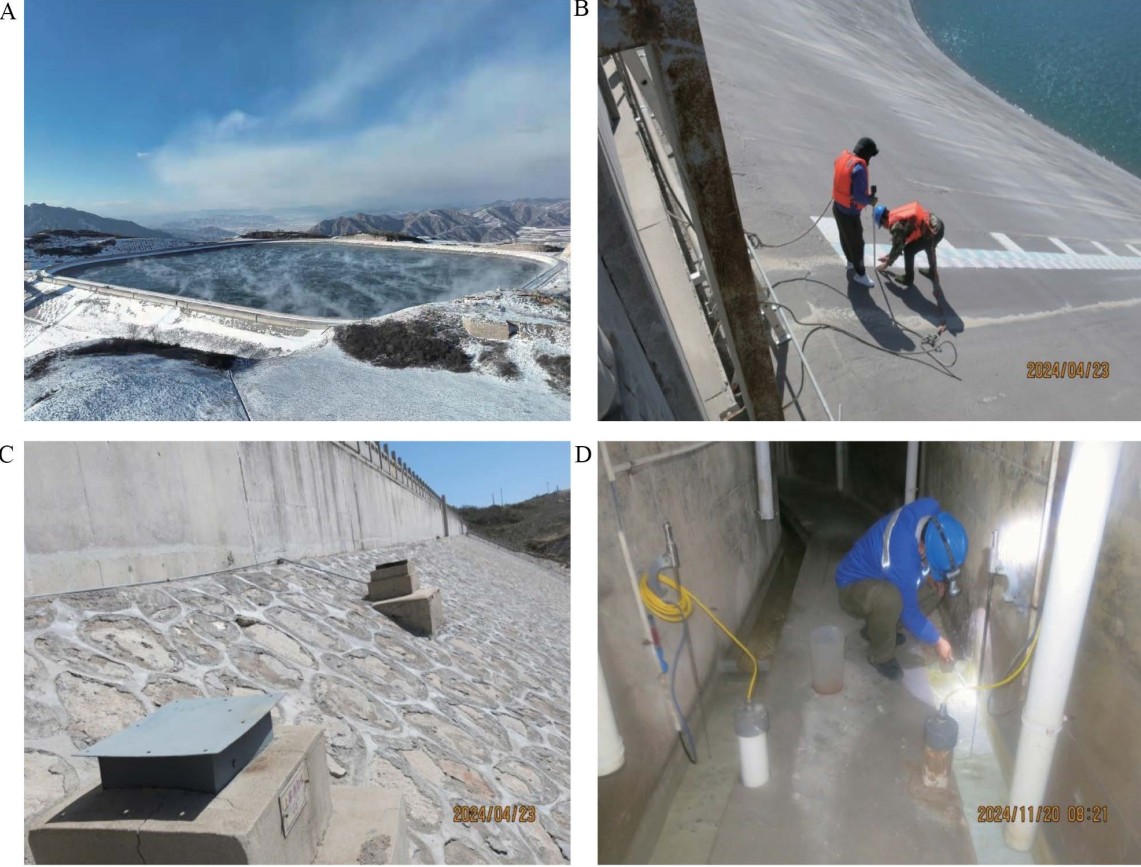

**Fig 1. Reservoir area and routine monitoring photographs.** (A) general map of the reservoir area, (B) daily monitoring of water-levels, (C) protective cover for sedimentation monitoring sites, (D) daily monitoring of seepage.

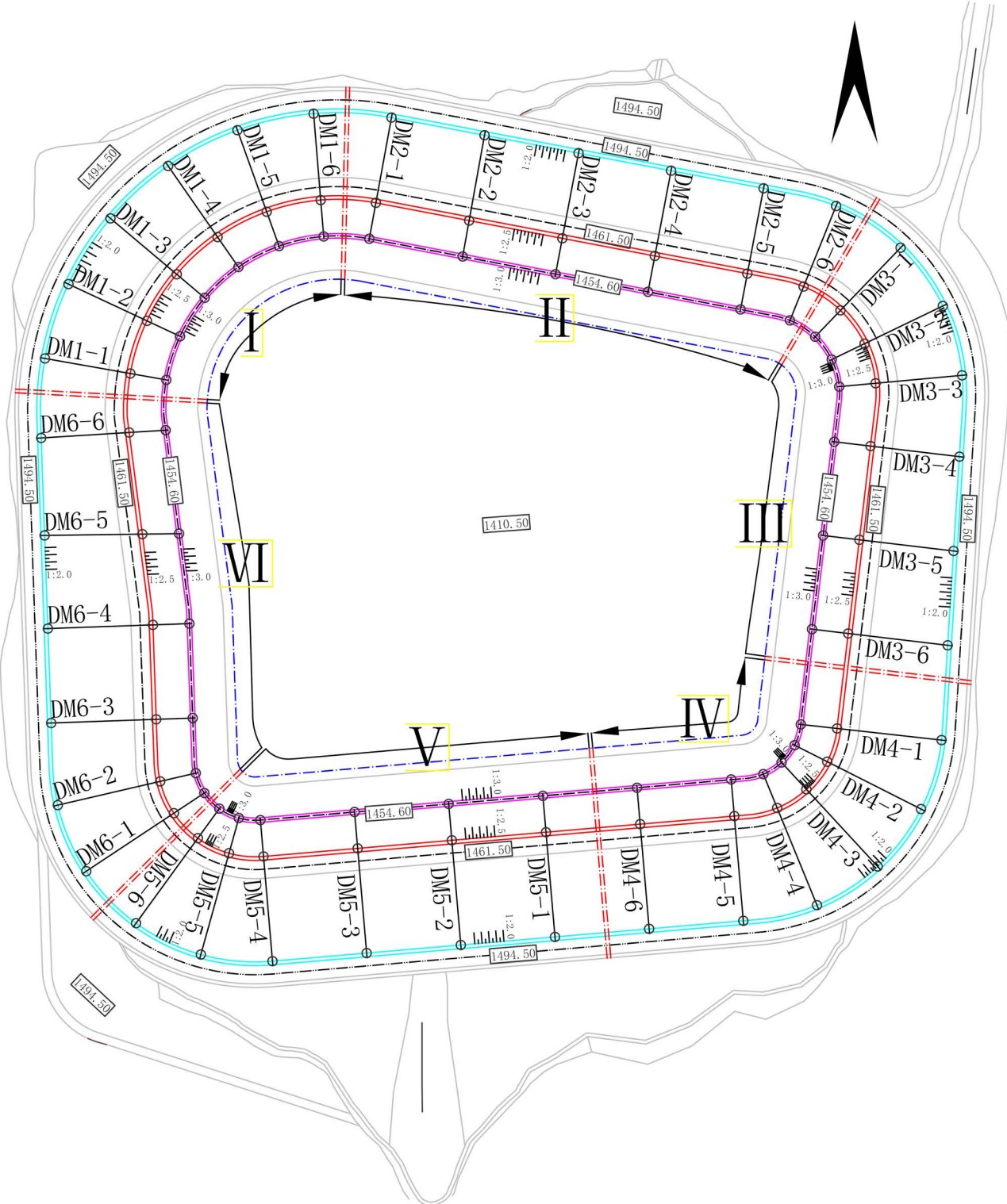

**Fig 2. Location of monitoring sites (Cyan solid line is top, red solid line is middle, magenta solid line is bottom, black solid line is section line, red dashed line is separation line, blue dashed line is water-level line).**

**Table 1. Distribution of cross sections in the reservoir area.**

| Area | Cross-sections | Place | Type of monitoring |
|---|---|---|---|
| I | DM1–1 DM1–2 DM1–3 DM1–4 DM1–5 DM1–6 | Top Middle Bottom | sedimentation(mm) fissure (mm) |
| II | DM2–1 DM2–2 DM2–3 DM2–4 DM2–5 DM2–6 | | tilt (mm) seepage(L/min) |
| III | DM3–1 DM3–2 DM3–3 DM3–4 DM3–5 DM3–6 | | soil-displacement(mm) stress (MPa) |
| IV | DM4–1 DM4–2 DM4–3 DM4–4 DM4–5 DM4–6 | | loading($10^3$kN) water-level(km) |
| V | DM5–1 DM5–2 DM5–3 DM5–4 DM5–5 DM5–6 | | |
| VI | DM6–1 DM6–2 DM6–3 DM6–4 DM6–5 DM6–6 | | |

total of 3,456 sets of measurements collected for the 36 sections. At the same time, in order to further understand the pattern of change of measurement data, the researchers through the plotting of different types of monitoring data 3D scatterplot for the preliminary analysis of the pattern of change (Fig 3). We conducted homogeneity of variance tests on monitoring data from different altitudes (top, middle, bottom) and different monitoring types within the same region. All tests showed significant violations of homogeneity ($p < 0.05$). We selected a significance level of $\alpha = 0.05$ (two-tailed) with a statistical power ($1-\beta$) of 0.95, and employed one-sample t-tests to estimate the required sample size. Following the initial analysis, rigorous post hoc testing was implemented through Welch's unequal variances t-test and Brown-Forsythe variance-robust ANOVA.

**2.2.3 Data integrity and reliability analysis.** In order to further examine the quality of the observed data, this study classified the observed data into 8 categories according to the type of data, and each category was divided into top, middle and bottom categories according to the location of the monitoring points. We plotted the above eight types of data as a boxplot with a normal distribution, and analyzed the upper and lower boundaries of the boxplot (upper quartile Q3 and lower quartile Q1), the median line (Q2), and the outliers. Meanwhile, the normal distribution curve superimposed on the box-and-line plot presents the general distribution trend of the data, and the normal curve has a mean of μ and a standard deviation of σ.

**2.2.4 Sensitive treatment of observational datasets.** We needed to normalize the observations within each region in order to know further before the sensitivity of the observations within each region [28,29]. The sample data of sedimentation, fissure, tilt, seepage, soil-displacement, stress, loading and water-level in areas I to VI were normalized in turn, and the calculation formula is as follows:

$$Y = \frac{X - X_{min}}{X_{max} - X_{min}}$$

(1)

Where $Y$ is the normalized value and $X, X_{max}, X_{min}$ are the observed values, maximum and minimum values for each category in turn.

**2.2.5 Inter-group variance analysis with post hoc tukey HSD.** The aim of this study was to investigate the effect of monitoring points being in different areas and different places on the observed data [30,31]. Measurements were repeated 4 times within each group with a total number of samples (n = 3456). In order to ensure the accuracy and consistency of the observed data, all operational steps during this study were carried out in accordance with are strictly in accordance with the specifications. Firstly, we analyzed intra-group correlations of observations within the regions by dividing them into three types, top, middle and bottom, according to their location; Then, we analyzed the intergroup correlation of

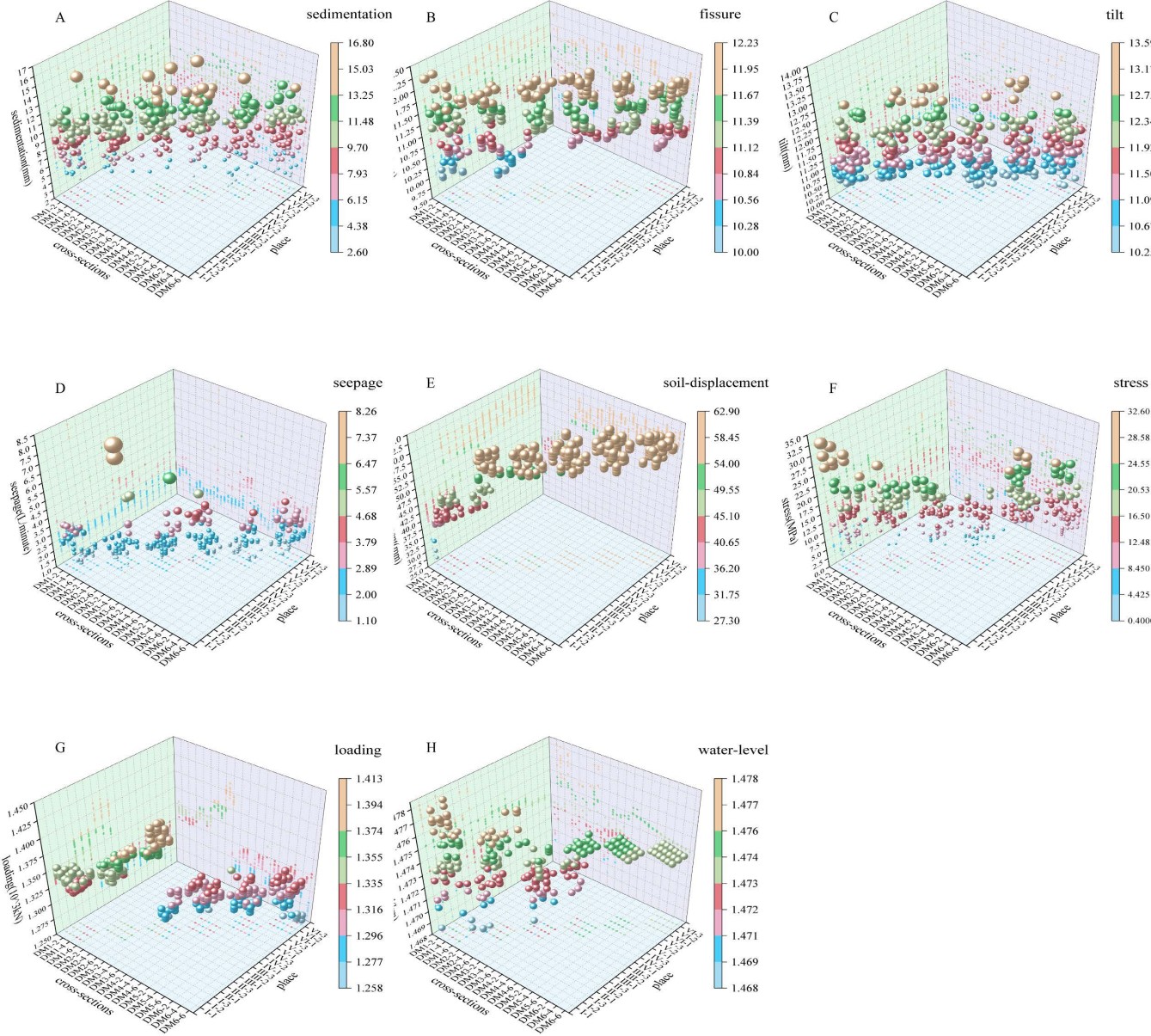

**Fig 3. 3D scatterplot of 8 types of monitoring data.** The horizontal axis is the section (DM indicates the section, and the number indicates the section number), the vertical axis is the area location (I to VI indicates the area I to VI in order, the number 1 is the top place, the number 2 is the middle place, and the number 3 is the bottom place), and the vertical axis is the axis of the change of the monitoring data. The size of the sphere represents the size of the measured value.

the observations between the six regions; Finally, we determined the strength of intra- and inter-group correlations of observations and the correlation coefficients between the eight observation types [4,5,31].

**2.2.6 Hierarchical clustering with correlation-based similarity.** The aim of this study was to identify the class groups of the observed data through correlation cluster analysis [32,33], which provided some reference for safety monitoring decision-making. reference for safety monitoring decision-making. We normalized the data in the region, chose hierarchical clustering, used Pearson's correlation coefficient as a similarity measure, and divided the observation types

into 2 groups and the observation sections into 4 groups. By drawing correlation clustering heatmaps for visual analysis, our study shows that the type of monitoring to different places makes a significant difference to the safety of the reservoir, this provides a valuable reference for making safety decisions in the future.

## 2.3 Statistical analysis

IBM SPSS Statistics 26 (SPSS Inc., USA) was used to analyze the observed data by ANOVA and correlation analysis, Welch/Brown-Forsythe post-hoc multiple tests were used in ANOVA, and two-tailed Pearson correlation was used in correlation analysis between groups to determine differences between groups, to assess the significance of the findings, the significance of the experimental results of this study was set at 0.05. We used the R environment (version: R-4.4.2, https://www.r-project.org/) to plot correlation clustering heatmaps [28,29]. Correlation heatmaps for clustering analysis of observations were plotted using the "pheatmap", "ggplot2", and "corrplot" packages in R, with significant differences of P<0.05, The results were visualized using the "ggplot2" package. Origin Pro 2024 (Origin Lab Inc., USA) was used to plot 2D and 3D scatter plots of the observations.

## 3 Results

### 3.1 Data quality analysis

To enable researchers, readers, and decision-makers to quickly understand the basic characteristics, distribution patterns, and underlying trends of the data, we performed descriptive statistical analysis on all types of monitoring data. Sedimentation data (n=432) ranged from 2.60 to 16.77 mm (mean±SD=9.42±2.51 mm) (Table 2). Fissure data (n=432) ranged from 10.0093 to 12.221 mm (mean±SD=11.3141±0.5356 mm) (Table 2). Lean data (n=432) ranged from 10.2523 to 13.5882 mm (mean±SD=11.3849±0.6484 mm) (Table 2). Seepage data (n=432) ranged from 1.1002 to 8.2451 L/min (mean±SD=2.4004±0.7803 L/min) (Table 2). Soil-displacement data (n=432) ranged from 27.3747 to 62.8343 mm (mean±SD=54.1826±6.7920 mm) (Table 2). Stress data (n=432) ranged from 0.4409 to 32.5349 MPa (mean±SD=14.0055±5.8074 MPa) (Table 2). Loading data (n=432) ranged from 1.2577 to 1.4129 103kN (mean±SD=1.3262±0.0325 103kN) (Table 2). Water-level data (n=432) ranged from 1.4680 to 1.4871 km (mean±SD=1.4738±0.0016 km) (Table 2).

A box-plot with a normal line is a statistical graph that describes the degree of dispersion of a set of data. The interquartile range *IQR* was used to measure the discrete degree of the data in the box-plot. the normal line is used to describe the distribution of the data.

From the sedimentation observation box-plot (Fig 4A), the minimum value of *IQR* occurs in the middle place was 3.22 mm, indicating that the middle place of the observation data was the least dispersed. From the fissure observation

**Table 2. Descriptive statistical analysis of monitoring data across different measurement types.**

| Monitoring Type | n | Max | Min | Mean | SD |
|---|---|---|---|---|---|
| Sedimentation(mm) | 432 | 2.6000 | 16.7700 | 9.4197 | ±2.5078 |
| Fissure (mm) | 432 | 10.0093 | 12.2210 | 11.3141 | ±0.5356 |
| Lean (mm) | 432 | 10.2523 | 13.5882 | 11.3849 | ±0.6484 |
| Seepage (L/min) | 432 | 1.1002 | 8.2451 | 2.4004 | ±0.7803 |
| Soil-displacement (mm) | 432 | 27.3747 | 62.8343 | 54.1826 | ±6.7920 |
| Stress (MPa) | 432 | 0.4409 | 32.5349 | 14.0055 | ±5.8074 |
| Loading (10³kN) | 432 | 1.2577 | 1.4129 | 1.3262 | ±0.0325 |
| Water-level (km) | 432 | 1.4680 | 1.4781 | 1.4738 | ±0.0016 |

Note : n denotes the sample size, Max denotes the maximum value, Min denotes the minimum value, Mean denotes the average value, SD denotes the standard deviation.

box-plot (Fig 4B), the minimum value of *IQR* occurs in the bottom place was 0.87 mm, indicating that the bottom place of the observation data is the least dispersed. From the tilt box-plot (Fig 4C), the minimum value of *IQR* occurs in the middle place was 0.59 mm, indicating that the middle place of the observation data was the least dispersed. From the seepage box-plot (Fig 4D), the minimum value of *IQR* occurs in the bottom place was 0.78 L/min, indicating that the bottom place of the observation data was the least dispersed, there were 2 large anomalies on top place, possibly due to errors in the recording of observations, the data were overwhelmingly shifted around the median, which didn't have a major impact on the results of the analysis. From the soil-displacement observation box-plot (Fig 4E), the minimum value of *IQR* occurs in the middle place was 5.71 mm, indicating that the middle place of the observation data was the least dispersed. From the stress box-plot (Fig 4F), the minimum value of *IQR* occurs in the middle place was 6.82 MPa, indicating that the middle place of the observation data was the least dispersed. From the loading box-plot (Fig 4G), the minimum value of

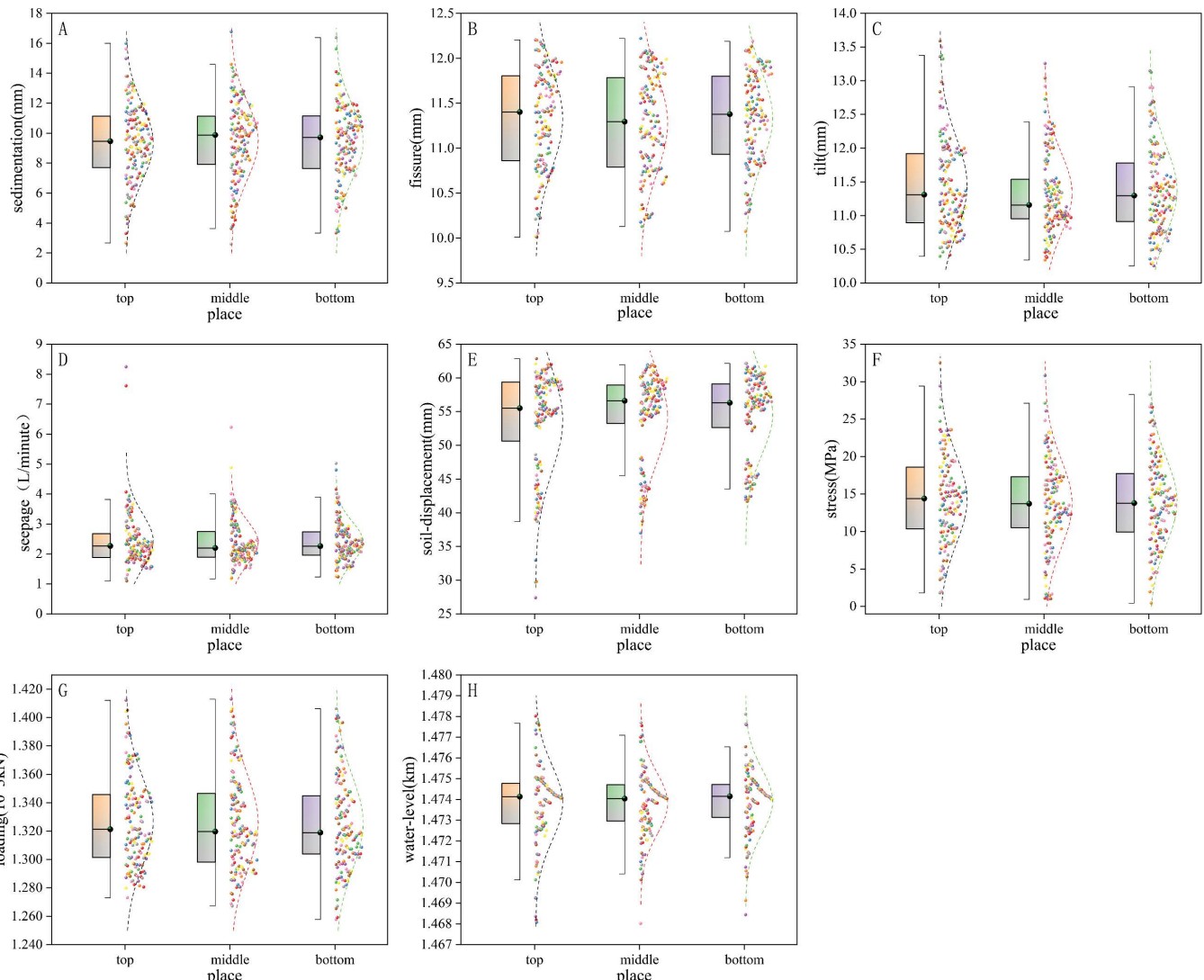

**Fig 4. Box-plots of different types of observations.** Variation of sedimentation at different place **(A)**, Variation of fissure at different place **(B)**, Variation of tilt at different place **(C)**, Variation of seepage at different place **(D)**, Variation of soil-displacement at different place **(E)**, Variation of stress at different place **(F)**, Variation of loading at different place **(G)**, Variation of water-level at different place **(H)**.

*IQR* occurs in the bottom place was 0.042 10^^3 KN, indicating that the bottom place of the observation data was the least dispersed. From the loading box-plot (Fig 4H), the minimum value of *IQR* occurs in the bottom place was 0.0016 km, indicating that the bottom place of the observation data was the least dispersed. Besides, the Kolmogorov-Smirnov test results indicated normal distributions (P > 0.05) for six of the eight monitoring data types across all 36 sections, with only seepage and water-level data demonstrating marginal normality (P > 0.05). The sample size estimation demonstrated that the required sample sizes for all monitored data categories were below the total sample size of 432. The robust tests for equality of means revealed statistically significant differences (P < 0.05) among: (1) different altitude levels (top/middle/bottom) within each monitoring type in the same region, and (2) distinct monitoring types within the identical geographical area.

## 3.2 Sensitivity analyses of observational data

A lollipop chart is a graph used to visual data and facilitate the presentation of comparisons between categories. It usually consists of individual points (representing data values) and vertical or horizontal lines connecting those points.

The length of the lollipop chart stick indicates the strength of sensitivity [18,20,24], the longer the stick the stronger the sensitivity and conversely the weaker the sensitivity (Fig 5). From the lollipop chart at area I (Fig 5A), observational sensitivity was highest for water-level observations and lowest for tilt observation at top place, observational sensitivity was highest for sedimentation observations and lowest for tilt observation at middle place, observational sensitivity was highest for fissure observations and lowest for soil-displacement observation at bottom place. From the lollipop chart at

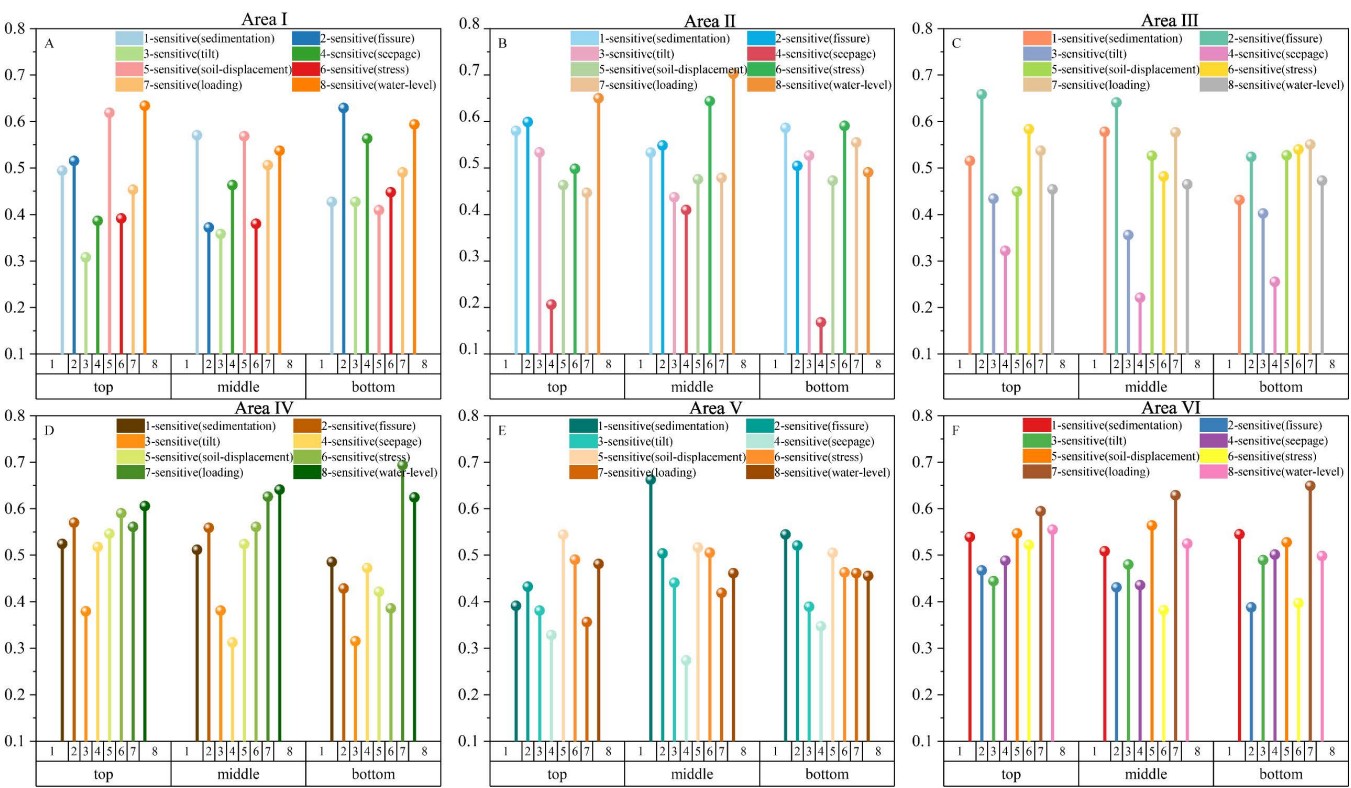

**Fig 5. Lollipop plots of the sensitivity of observations at different place.** Note: A, B, C, D, E, and F, correspond, respectively, to Area I, Area II, Area III, Area IV, Area V, Area VI. 1, 2, 3, 4, 5, 6, 7 and 8 represent the sensitivity to sedimentation, fissure, tilts, seepage, soil-displacement, stress, loading and water-level respectively.

area II (Fig 5B), observational sensitivity was highest for water-level observations and lowest for seepage observation at top place, observational sensitivity was highest for water-level observations and lowest for seepage observation at middle place, observational sensitivity was highest for sedimentation observations and lowest for seepage observation at bottom place. From the lollipop chart at area III (Fig 5C), observational sensitivity was highest for fissure observations and lowest for seepage observation at top place, observational sensitivity was highest for fissure observations and lowest for seepage observation at middle place, observational sensitivity was highest for loading observations and lowest for seepage observation at bottom place. place. From the lollipop chart at area IV (Fig 5D), observational sensitivity was highest for water-level observations and lowest for tilt observation at top place, observational sensitivity was highest for water-level observations and lowest for seepage observation at middle place, observational sensitivity was highest for loading observations and lowest for tile observation at bottom place. From the lollipop chart at area V (Fig 5E), observational sensitivity was highest for soil-displacement observations and lowest for seepage observation at top place, observational sensitivity was highest for sedimentation observations and lowest for seepage observation at middle place, observational sensitivity was highest for seepage observations and lowest for seepage observation at bottom place. From the lollipop chart at area VI (Fig 5F), observational sensitivity was highest for loading observations and lowest for tilt observation at top place, observational sensitivity was highest for loading observations and lowest for stress observation at middle place, observational sensitivity was highest for loading observations and lowest for fissure observation at bottom place.

### 3.3 significance analysis

**3.3.1 Significance analysis of observations from different place within the same area.** To understand the significance of observations from different place within the same area [23,34], we performed one-way ANOVA Least Significant Difference (LSD) and Tukey Honest Significant Difference (HSD) post-hoc multiple comparisons of eight categories of observations within the same place (Table 3 and Fig 5).

In area I, the maximum mean values of sedimentation, fissure, tilts, seepage, soil-displacement, stress, loading and water-level were maximum $8.7908 \pm 0.3899$ mm, $11.241 \pm 0.1143$ mm, $11.4817 \pm 0.1314$ mm, $2.4538 \pm 0.1241$ L/min, $43.9636 \pm 0.3099$ mm, $15.9253 \pm 1.1892$ MPa, $1.3369 \pm 0.0014$ $10^3$ kN, $1.4745 \pm 0.0005$ km, respectively; On the top sedimentation, soil-displacements and stresses were significantly higher ($p < 0.05$); On the middle sedimentation and soil-displacements were significantly higher ($p < 0.05$); On the bottom sedimentation, soil-displacements and stress were significantly higher ($p < 0.05$) (Fig 6A).

In area II, the maximum mean values of sedimentation, fissure, tilts, seepage, soil-displacement, stress, loading and water-level were maximum $10.1767 \pm 0.4726$ mm, $11.2403 \pm 0.1285$ mm, $11.8247 \pm 0.1428$ mm, $2.6439 \pm 0.3399$ L/min, $52.0574 \pm 0.9946$ mm, $15.186 \pm 1.0641$ MPa, $1.3592 \pm 0.0029$ $10^3$ kN, $1.4746 \pm 0.0003$ km, respectively; On the top sedimentation, soil-displacements and stresses were significantly higher ($p < 0.05$); On the middle soil-displacements and stress were significantly higher ($p < 0.05$); On the bottom sedimentation, soil-displacements and stress were significantly higher ($p < 0.05$) (Fig 6B).

In area III, the maximum mean values of sedimentation, fissure, tilts, seepage, soil-displacement, stress, loading and water-level were maximum $10.7479 \pm 0.4669$ mm, $11.6744 \pm 0.0625$ mm, $11.8969 \pm 0.1409$ mm, $2.4461 \pm 0.1306$ L/min, $56.3058 \pm 0.4582$ mm, $15.4428 \pm 1.2328$ MPa, $1.355 \pm 0.0100$ $10^3$ kN, $1.4730 \pm 0.0004$ km, respectively; On the top soil-displacements and stress were significantly higher ($p < 0.05$); On the middle soil-displacements and stresses were significantly higher ($p < 0.05$); On the bottom sedimentation, seepage, soil-displacements and stress were significantly higher ($p < 0.05$) (Fig 6C).

In area IV, the maximum mean values of sedimentation, fissure, tilts, seepage, soil-displacement, stress, loading and water-level were maximum $10.8454 \pm 0.4704$ mm, $11.5539 \pm 0.1019$ mm, $11.1523 \pm 0.1144$ mm, $2.7560 \pm 0.1660$ L/min, $58.1181 \pm 0.4437$ mm, $9.9275 \pm 1.2078$ MPa, $1.3081 \pm 0.0024$ $10^3$ kN, $1.4739 \pm 0.0003$ km,

**Table 3. ANOVA (Duncan's multiple range test) changes in type of observation at different place within the same area.**

| area | place | Sedimentation (mm) | Fissure (mm) | Tilt (mm) | Seepage (L/min) | Soil-displacement(mm) | Stress (MPa) | Loading ($10^3$kN) | Water-level (km) |
|---|---|---|---|---|---|---|---|---|---|
| I | top | 8.4996±0.3452d | 11.1393±0.1341c | 11.0909±0.0793c | 2.4538±0.1241e | 40.5271±1.1561a | 14.9142±1.6750b | 1.3391±0.0014e | 1.4745±0.0005e |
| | middle | 8.7908±0.3899c | 10.907±0.1282b | 11.3373±0.1225b | 2.3917±0.0981d | 42.4398±0.5448a | 12.3793±1.6119b | 1.3349±0.0014d | 1.4742±0.0005d |
| | bottom | 8.5850±0.5444d | 11.2410±0.1143c | 11.4817±0.1314c | 2.3050±0.1063e | 43.9636±0.3099a | 15.9253±1.1892b | 1.3369±0.0014e | 1.4742±0.0005e |
| II | top | 8.9350±0.3270d | 11.2403±0.1285c | 11.4820±0.1096c | 2.6439±0.3399e | 50.8283±1.3217a | 14.6803±1.2730b | 1.3540±0.0025e | 1.4731±0.0005e |
| | middle | 10.1767±0.4726c | 11.1448±0.1244c | 11.0869±0.0552c | 2.2270±0.1202d | 51.2387±1.2800a | 15.1860±1.0641b | 1.3576±0.0028d | 1.4735±0.0004d |
| | bottom | 9.3075±0.4375d | 11.1084±0.1171c | 11.8247±0.1428c | 2.3981±0.1603e | 52.0574±0.9946a | 13.7524±0.9549b | 1.3592±0.0029e | 1.4746±0.0003e |
| III | top | 10.7479±0.4669c | 11.4524±0.1047c | 11.8969±0.1409c | 2.2271±0.07300d | 56.0148±0.5128a | 15.4428±1.2328b | 1.3520±0.0098d | 1.4730±0.0004d |
| | middle | 10.6433±0.4040c | 11.5309±0.0918bc | 11.5424±0.1481bc | 2.3073±0.1895d | 56.3058±0.4582a | 12.6229±0.9213b | 1.3550±0.0100d | 1.4729±0.0002d |
| | bottom | 10.4700±0.4533d | 11.6744±0.0625c | 11.6582±0.1155c | 2.4461±0.1306e | 56.2044±0.4563a | 13.7526±0.6489b | 1.3488±0.0096f | 1.4724±0.0003f |
| IV | top | 10.8454±0.4704bc | 11.5539±0.1019b | 11.1523±0.1144b | 2.6361±0.1680d | 57.6005±0.4718a | 9.9170±0.8067c | 1.3077±0.0022e | 1.4738±0.0003e |
| | middle | 10.3542±0.5886bc | 11.4919±0.0999b | 11.1203±0.1132bc | 2.7560±0.1660d | 57.6431±0.4316a | 9.9275±1.2078c | 1.3081±0.0024d | 1.4737±0.0003d |
| | bottom | 10.5375±0.3798b | 11.4075±0.1058b | 11.1090±0.1346b | 2.5549±0.1604d | 58.1181±0.4437a | 7.7501±0.9554c | 1.3072±0.0026e | 1.4739±0.0003e |
| V | top | 7.7362±0.6717d | 11.3095±0.0845c | 11.6914±0.2030c | 2.0719±0.1320e | 58.1181±0.4702a | 16.4005±0.8139b | 1.3015±0.0028e | 1.4745±0.0000e |
| | middle | 9.7504±0.4885d | 11.3528±0.0772c | 11.3317±0.1148c | 2.0865±0.1329e | 58.5271±0.4514a | 16.9211±0.7458b | 1.2984±0.0025e | 1.4744±0.0000e |
| | bottom | 8.3692±0.5378d | 11.4219±0.0818c | 11.0310±0.0880c | 2.0813±0.1115e | 58.6962±0.4820a | 17.9799±1.0739b | 1.3004±0.0026e | 1.4744±0.0000e |
| VI | top | 9.4300±0.4168d | 11.2638±0.1041c | 11.3765±0.1177c | 2.5567±0.1577e | 59.1702±0.5680a | 14.3235±0.7383b | 1.3070±0.0032e | 1.4741±0.0000e |
| | middle | 7.2921±0.4928d | 11.2293±0.1034c | 11.5747±0.1276c | 2.4934±0.1432e | 59.1906±0.3838a | 15.2043±0.9129b | 1.3016±0.0038f | 1.4740±0.0000ef |
| | bottom | 9.0833±0.5090d | 11.1846±0.0901c | 11.1408±0.0874c | 2.5706±0.1394e | 58.6428±0.3962a | 15.0185±1.0723b | 1.3016±0.0043e | 1.4742±0.0000e |

Note: Values are presented as mean±standard error. Same superscripts in a row are not significantly different (p > 0.05) [29,32].

respectively; On the top seepage and soil-displacements were significantly higher (p < 0.05); On the middle fissure and soil-displacements were significantly higher (p < 0.05); On the bottom seepage and soil-displacements were significantly higher (p < 0.05) (Fig 6D).

In area V, the maximum mean values of sedimentation, fissure, tilts, seepage, soil-displacement, stress, loading and water-level were maximum 9.7504±0.4885 mm, 11.4219±0.0818 mm, 11.6914±0.2030 mm, 2.0865±0.1329 L/min, 58.6962±0.4820 mm, 17.9799±1.0739 MPa, 1.3015±0.0028 $10^3$ kN, 1.4745±0.0000 km, respectively; On the top sedimentation, soil-displacements and stress were significantly higher (p < 0.05); On the middle sedimentation, soil-displacements and stress were significantly higher (p < 0.05); On the bottom sedimentation, soil-displacements and stress were significantly higher (p < 0.05) (Fig 6E).

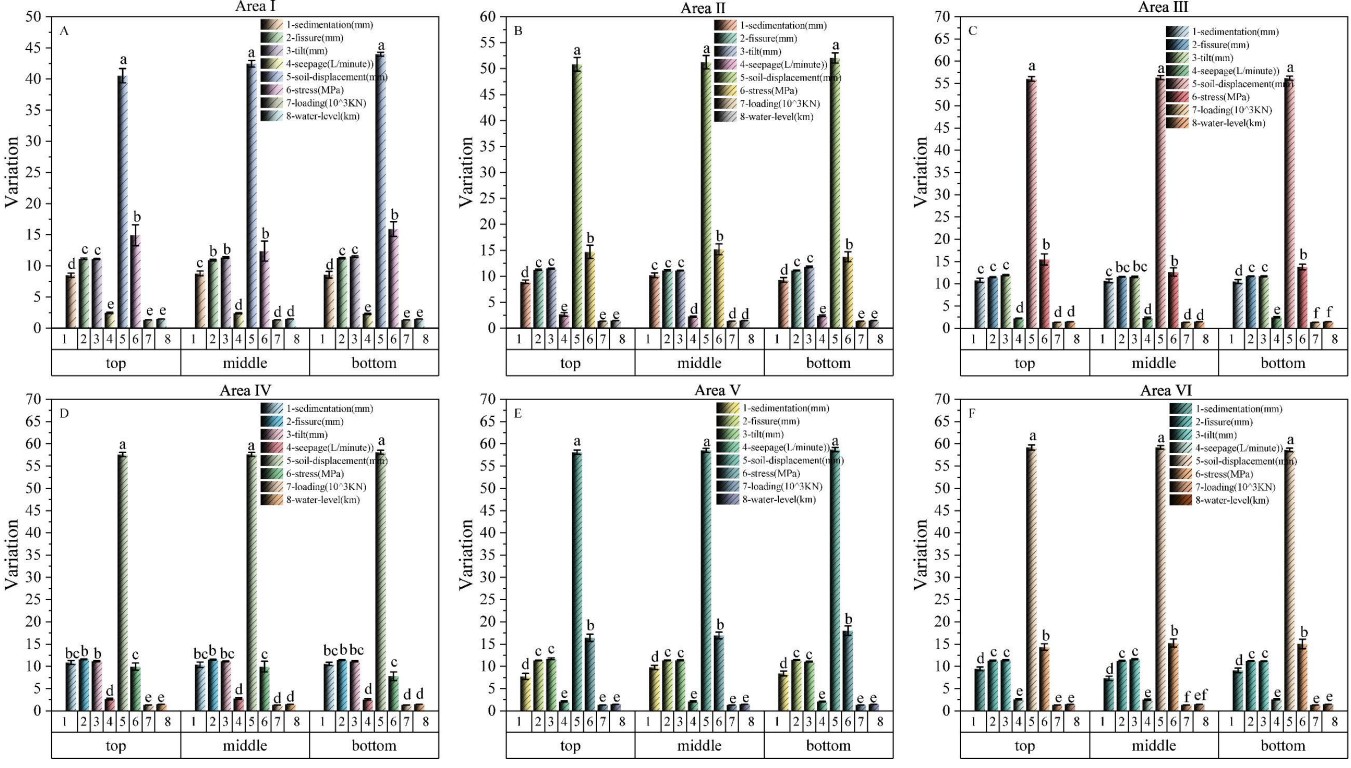

**Fig 6. Impacts between different observation types at different place within the same area.** The bar graph represents the mean, error bars represent the standard deviation and different letters indicate significant difference at p < 0.05. Note: A, B, C, D, E, and F, correspond, respectively, to Area I. Area II, Area III, Area IV, Area V, Area VI. 1, 2, 3, 4, 5, 6, 7 and 8 represent observations of sedimentation, fissure, tilts, seepage, soil-displacement, stress, loading and water-level respectively.

In area VI, the maximum mean values of sedimentation, fissure, tilts, seepage, soil-displacement, stress, loading and water-level were maximum 9.4300 ± 0.4168 mm, 11.2638 ± 0.1041 mm, 11.5747 ± 0.1276 mm, 2.5706 ± 0.1394 L/min, 59.1906 ± 0.3838 mm, 15.2043 ± 0.9129 MPa, 1.3070 ± 0.0032 $10^3$ kN, 1.4742 ± 0.0000 km, respectively; On the top sedimentation, soil-displacements and stress were significantly higher (p < 0.05); On the middle sedimentation, soil-displacements, seepage and stress were significantly higher (p < 0.05); On the bottom sedimentation, soil-displacements and stress were significantly higher (p < 0.05) (Fig 6F).

**3.3.2 Significance analysis of observations from different area.** To further clarify the significance of the observations in different regions, we performed one-way ANOVA LSD and Tukey HSD post hoc multiple comparisons of the number of observations of the same type in the six areas (Table 4 and Fig 7).

From sedimentation data, we knew that the maximum average value was 10.6204 ± 0.2521 mm for area III; From fissure data, we knew that the maximum average value was 11.5526 ± 0.0513 mm for area III; From tilt data, we knew that the maximum average value was 11.6992 ± 0.0791 mm for area III; From seepage data, we knew that the maximum average value was 2.6490 ± 0.0943 L/min for area IV; From soil-displacement data, we knew that the maximum average value was 59.0012 ± 0.2619 mm for area VI; From stress data, we knew that the maximum average value was 17.1005 ± 0.5121 MPa for area V; From loading data, we knew that the maximum average value was 1.3569 ± 0.0016 $10^3$kN for area II; From water-level data, we knew that the maximum average value was 1.4744 ± 0.0000 Km for area V.

Sedimentation, soil-displacement and stress were significantly higher (p < 0.05), in area I, II, III, V, VI. fissure, soil-displacement and stress were significantly higher (p < 0.05), in area IV.

**Table 4. ANOVA (Duncan's multiple range test) changes in type of observation at different area.**

| area | Sedimentation (mm) | Fissure (mm) | Tilt (mm) | Seepage (L/min) | Soil-displacement (mm) | Stress (MPa) | Loading ($10^3$kN) | water-level (km) |
|---|---|---|---|---|---|---|---|---|
| I | 8.6251± 0.2480d | 11.0958± 0.0735c | 11.3033± 0.0673c | 2.3835± 0.0630e | 42.3101± 0.4632a | 14.4063± 0.8761b | 1.3370± 0.0008e | 1.4743± 0.0003e |
| II | 9.4731± 0.2452d | 11.1645± 0.0706c | 11.4645± 0.0715c | 2.4230± 0.1312e | 51.3748± 0.6900a | 14.5396± 0.6330b | 1.3569± 0.0016e | 1.4738± 0.0002e |
| III | 10.6204± 0.2521d | 11.5526± 0.0513c | 11.6992± 0.0791c | 2.3268± 0.0801e | 56.1750± 0.2716a | 13.9394± 0.5658b | 1.3519± 0.0056f | 1.4728± 0.0002ef |
| IV | 10.5790± 0.2783c | 11.4844± 0.0588b | 11.1272± 0.069bc | 2.6490± 0.0943e | 57.7872± 0.2573a | 9.1982± 0.5841d | 1.3077± 0.0014f | 1.4738± 0.0002f |
| V | 8.6186± 0.3401d | 11.3614± 0.0466c | 11.3513± 0.0880c | 2.0799± 0.0716e | 58.4471± 0.2679a | 17.1005± 0.5121b | 1.3001± 0.0015f | 1.4744± 0.0000ef |
| VI | 8.6018± 0.2921d | 11.2259± 0.0567c | 11.3640± 0.0673c | 2.5402± 0.0838e | 59.0012± 0.2619a | 14.8488± 0.5244b | 1.3034± 0.0022e | 1.4740± 0.0000e |

Note: Values are presented as mean±standard error. Same superscripts in a row are not significantly different (p > 0.05) [29,32].

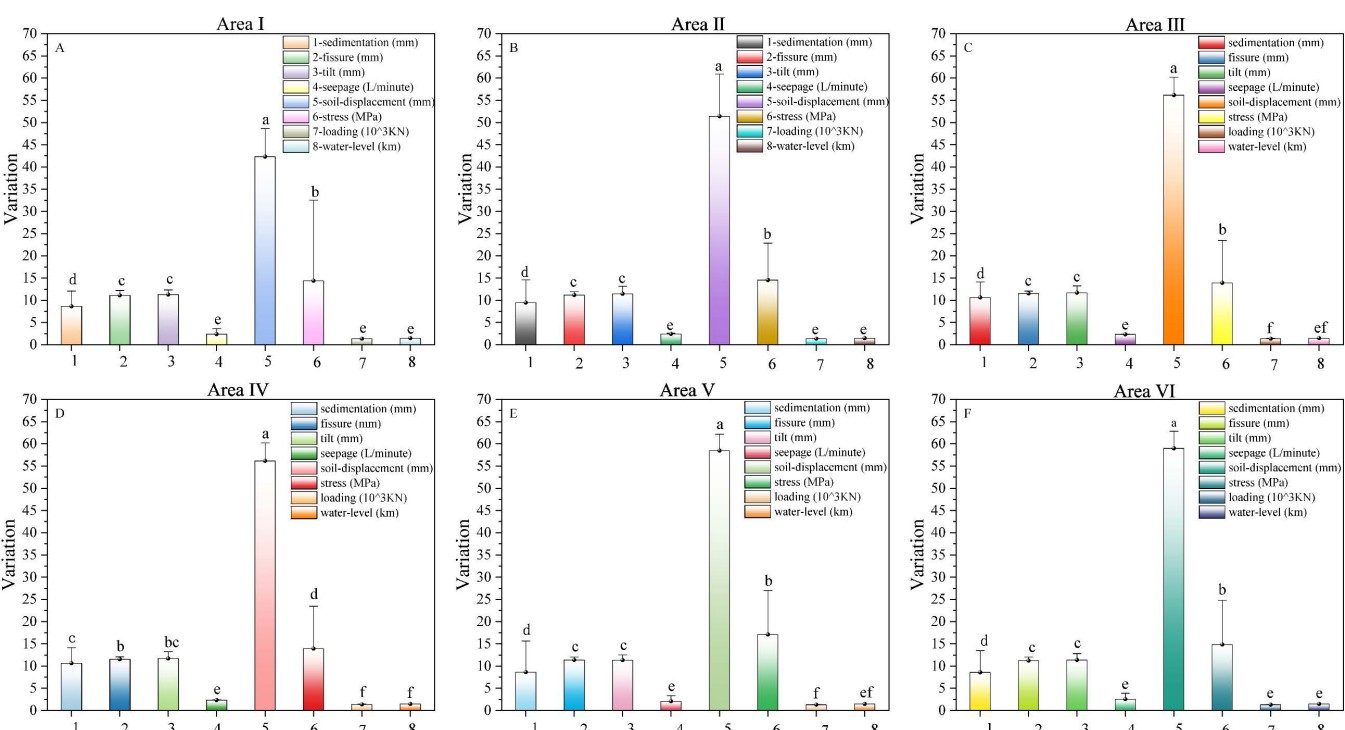

**Fig 7. Impacts between different observation types at different area.** The bar graph represents the mean, error bars represent the standard deviation and different letters indicate significant difference at p < 0.05. Note: A, B, C, D, E, and F, correspond, respectively, to Area I. Area II, Area III, Area IV, Area V, Area VI. 1, 2, 3, 4, 5, 6, 7 and 8 represent observations of sedimentation, fissure, tilts, seepage, soil-displacement, stress, loading and water-level respectively.

### 3.4 Pearson's correlation analysis and Mantel's test

To further explore the complex relationship between the different place (top, middle, and bottom) and several variables such as the type of observation in this study, multivariate statistical analyses.

It was clear that there was a strong correlation between soil-displacement and where the monitoring points were located (top, middle and bottom). Stress was correlated with where the monitoring point was placed (top and middle), it was weakly correlated with bottom. Sedimentation, fissure, tilt, seepage, loading and water-level all correlate weakly with where the monitoring points were located (top, middle, and bottom) (Table 5 and Fig 8).

Stress was significantly negatively correlated with sedimentation and seepage, respectively, while stress was also claimed to be significantly negatively correlated with soil-displacement, loading was significantly negatively correlated with tilt (Table 5 and Fig 8).

### 3.5 Hierarchical cluster analysis of different place observation types

In order to study more intuitively the effect of different place on different observation types of data, we further performed correlation clustering heatmap analyses for observations that were at 24 different place (Fig 9).

As could be seen from the types of observations, soil-displacement, stress and water-level were grouped together in one category; loading, seepage, fissure, sedimentation and tilt are grouped in another (Fig 9).

Further, the location of the observation type could be seen to fall into 4 broad categories. T (soil-displacement, M (soil-displacement, and B (soil-displacement) were first category; T (stress), M (stress), B (stress), T (water-level), M (water-level), and B (water-level) were the second category; B (tilt), T (loading), M (loading), and B (loading) were the third category; T (seepage), M (seepage), B (seepage), T (tilt), M (tilt), T (sedimentation), M (sedimentation), B (sedimentation), T (fissure), M (fissure) and B (fissure) were the fourth category.

## 4 Discussion

The safety of the reservoir area heavily relied on the reliability and accuracy of monitoring data [11,35–39]. Our results demonstrated that seepage changes were less significant than other factors (Fig. 3), aligning with their low sensitivity in sensitivity analyses (Fig. 4). In addition, the results revealed spatially varying correlations among monitoring types. Specifically, soil-displacement and stress exhibited a strong correlation, whereas no significant relationships were observed between sedimentation, fissure, and tilt or among seepage, loading, and water-level (Figs 6–7).

This study demonstrates that soil-displacement and stress variations exhibit strong spatial dependence. Additionally, stress showed significant negative correlations with seepage and soil-displacement, while loading was negatively

**Table 5. Pearson correlation coefficients (r) and corresponding significance p-values among different monitoring types are presented.**

|  | sedimentation | fissure | tilt | seepage | soil-displacement | stress | loading | water-level |
|---|---|---|---|---|---|---|---|---|
| sedimentation | – | 0.20 | 0.13 | 0.04 | 0.18 | **-0.39** | 0.20 | -0.33 |
| fissure | 0.25 | – | 0.15 | -0.13 | 0.11 | -0.17 | -0.02 | -0.30 |
| tilt | 0.43 | 0.39 | – | -0.10 | 0.14 | 0.05 | **0.40** | -0.22 |
| seepage | 0.84 | 0.46 | 0.55 | – | 0.01 | **-0.34** | 0.02 | -0.15 |
| soil-displacement | 0.30 | 0.51 | 0.40 | 0.97 | – | -0.08 | **-0.36** | 0.09 |
| stress | **0.02** | 0.31 | 0.75 | **0.04** | 0.64 | – | 0.06 | 0.07 |
| loading | 0.25 | 0.89 | **0.02** | 0.89 | **0.03** | 0.74 | – | -0.24 |
| water-level | 0.05 | 0.07 | 0.20 | 0.39 | 0.62 | 0.69 | 0.15 | – |

Note: Upper triangle displays Pearson's correlation coefficients (r), while the lower triangle shows Spearman's rank correlations (ρ). Significance levels: *p<0.05, ***p<0.01, *p<0.001 (two-tailed). Diagonal entries ('-') represent self-correlations.

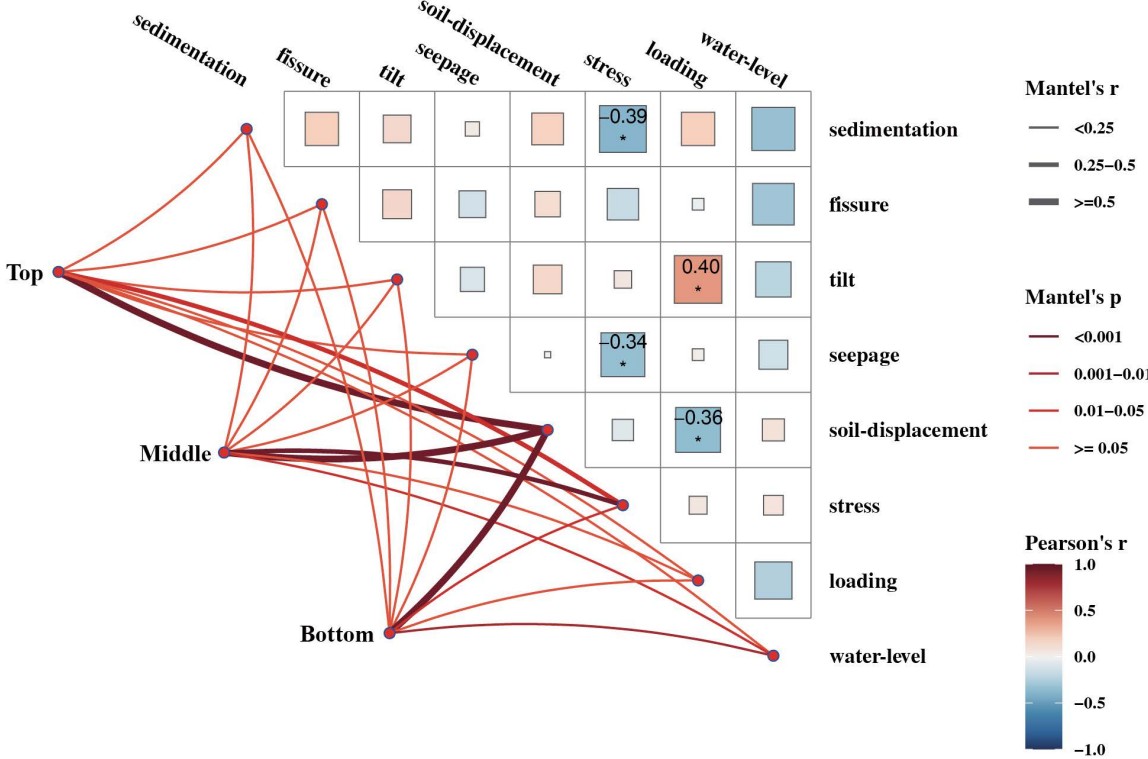

**Fig 8. Pearson's correlation analysis and Mantel's test.** Correlation heatmap Pearson correlation analysis is done for each monitoring type on the right, and the colors indicate the correlation coefficients. Blue and red squares indicate positive and negative correlation between two variables respectively [29,32]. Squares in red and blue colors show, respectively, positive and negative relationships between two variables. Significant correlations are indicated by *p<0.05, **p<0.01, ***p<0.001 [28,29,33]. The place was correlated with the type of monitoring through mental Mantel's test. In the left panel, the points represent monitoring sites categorized into three levels (top, middle, bottom) and analyzed via Mantel's test. The width and color of the connecting lines in the middle section denote the r statistic and significance p-value, respectively.

correlated with soil-displacement but positively correlated with tilt (Fig 7) [40,41]. Furthermore, the monitoring systems could be classified into two main categories (8 types) and four broader categories (24 groups) based on their spatial distribution (Fig 8).

Our monitoring data followed a normal distribution [42,43], except for seepage, which exhibited sporadic outliers potentially attributable to observer recording errors (Fig 2D). Moreover, our distribution results align with the safety studies by Huang et al. [5,10], which highlighted the high sensitivity of water-level changes to reservoir safety [18,20,24]. The conclusion of this study regarding the limited impact of seepage on reservoir bank stability aligns with the findings of Jia et al. (2024) [12], who demonstrated that multi-fracture seepage effects progressively diminish as reservoir water depth approaches groundwater-levels. Our reservoir system, characterized by a maximum water-level of 1492.5 m, minimum level of 1410.5 m, and an average depth of 42 m (Fig 1), falls within the classification of deep hydraulic structures. Furthermore, the monitoring data presented in Table 5 and Fig 7 confirm statistically insignificant correlations between seepage dynamics and critical deformation parameters (slope sedimentation: $R^2=0.0016$, $p=0.84$; inclination variation: $R^2=0.01$, $p=0.55$), reinforcing the conclusion of negligible seepage influence. This study revealed correlations among various monitoring data types, consistent with findings by Tang et al. [5,10]. Additionally, soil-displacement showed significant spatial variation (top/middle/bottom positions), attributable to year-round water storage in the reservoir area, which induced slope soil loosening. Our findings on monitoring-type clustering aligned with Vassileva et al.'s classification

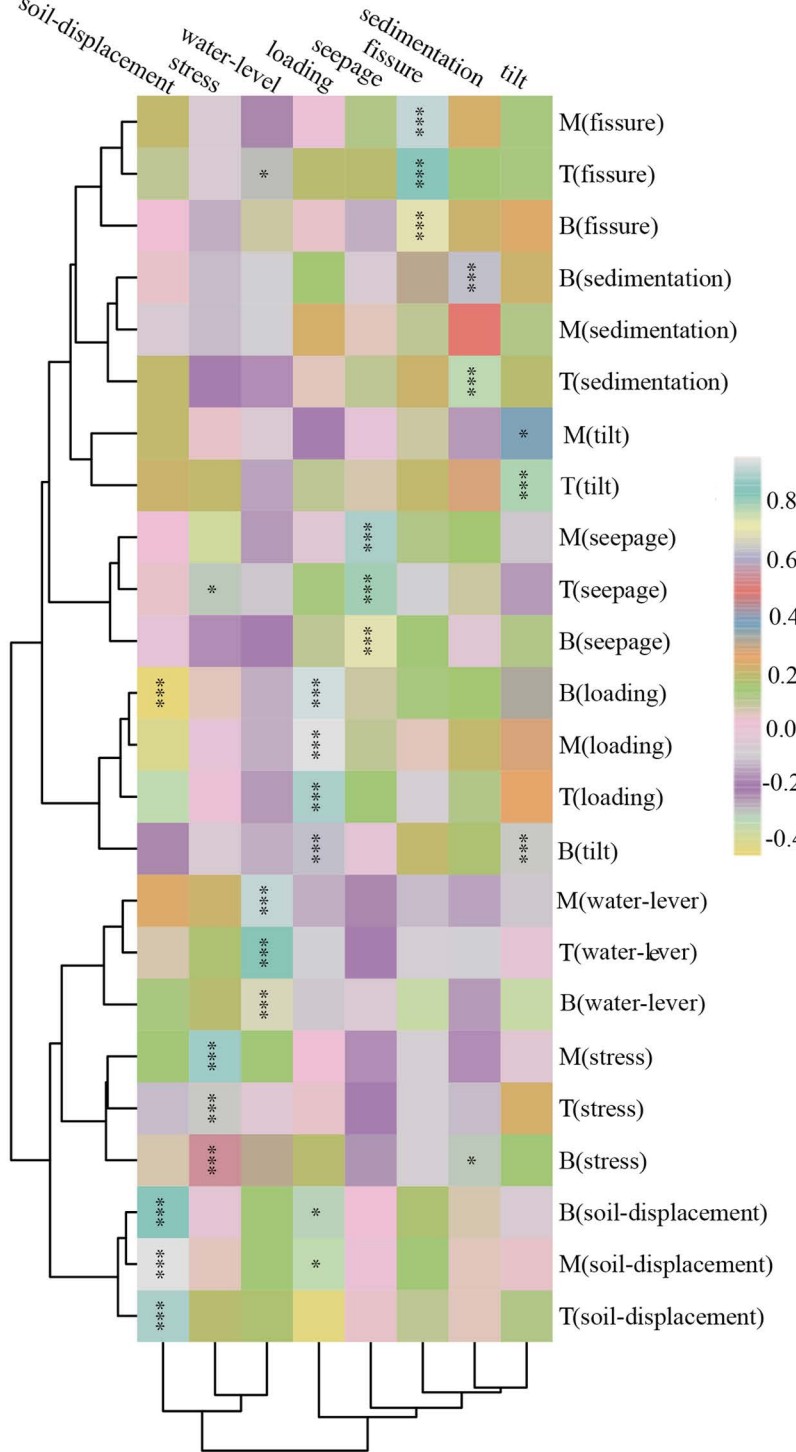

**Fig 9. Hierarchical cluster analysis heatmap of correlations for different place observation types.** We performed correlation-based cluster analysis on monitoring data across categories and locations (top, middle, bottom). Statistical significance is denoted by *p<0.05, **p<0.01, and ***p<0.001 [28,29,33]. In the figure: The capital letters T, M, and B preceding monitoring categories on the right indicate their respective positions (top, middle, bottom). Left-side lines depict clustering patterns between monitoring categories at distinct positions. Bottom lines illustrate clustering relationships among monitoring categories themselves.

of external and internal monitoring [9,11,43]. However, loading and seepage were misclassified as external monitoring despite belonging to internal monitoring. This discrepancy likely stems from correlated behavior among seepage, sedimentation, fissure, and tilt under sustained loading conditions.

The recommendations for reservoir area safety management primarily encompass the following measures: (1) implementation of real-time monitoring and surveillance systems to ensure continuous data acquisition; (2) enhancement of routine safety inspections and patrols through intensified frequency and systematic documentation; (3) establishment of a three-tiered emergency response mechanism (alert/warning/action levels) specifically designed for safety monitoring scenarios; and (4) development of a dynamic risk assessment framework that integrates real-time monitoring data with analytical protocols for timely risk identification and mitigation strategies.

Our findings differ notably from previous studies by Tang et al. (2019) and Kuang et al. (2024), which reported strong positive correlations between water-level fluctuations and slope sedimentation in the Three Gorges Reservoir area, as well as from Wang et al. (2023), who identified a negative correlation at the Xiaolangdi Dam [10,42]. In contrast, we not only quantified the magnitude of correlations but also explicitly analyzed their directions between reservoir water-levels and multiple monitoring parameters. Specifically, slope sedimentation, fissure, tilt, seepage, and loading all exhibited negative correlations with water-levels ($r = -0.33$, $-0.30$, $-0.22$, $-0.15$, and $-0.24$, respectively). Conversely, weak positive correlations were observed between water-level changes and soil-displacement/stress ($r = 0.09$ and $0.07$). These results further elucidate the mechanistic interplay between reservoir operations and slope behavior (Table 5).

The primary strength of this study lies in the systematic categorization and statistical analysis of monitoring data [11,44], enabling identification of key factors influencing reservoir safety [45,46]. These findings provide valuable insights for managerial decision-making [30,37]. At the same time, our study has proved that changes in water-level, sedimentation, fissure and soil-displacement play a vital role in the safety of reservoirs, and we should strengthen our monitoring efforts in this area.

Our results further confirm that internal slope changes influence external deformations. These findings advance understanding of reservoir security mechanisms and could be extended to other applications, particularly high-slope safety monitoring.

This study has two main limitations: (1) the relatively small sample size may affect the generalizability of our findings, and (2) the exclusive focus on impoundment conditions does not address reservoir safety during water release.

Our next steps in this research will be to conduct cross-reservoir validation studies to assess the generalizability of identified relevant models, multifactorial analyses that incorporate correlations of changes in each type of monitoring data during reservoir releases, and the implementation of an automated slope monitoring system. The integrated framework's goal is to use predictive analysis to connect monitoring data to operational hazards in reservoir project management.

## 5 Conclusion

In summary, we have demonstrated that there is correlation and clustering between the individual monitoring types on reservoir slopes, and that the clustering between the monitoring types is related to their location, indicating that changes in the monitoring data play a crucial role in the safety of the reservoir.

In conclusion, the results of this study found that seepage was the least sensitive to the security aspects of the reservoir. In terms of sensitivity of monitoring types, there is a significant correlation between soil-displacement, stress and others, in addition, stress was significantly negatively correlated with sedimentation and soil-displacement, and loading was also significantly negatively correlated with soil-displacement, while loading was positively correlated with tilt, meanwhile, the soil-displacement at different place is significantly changed. Moreover, cluster analyses revealed that the monitoring types could be divided into 2 major categories among themselves and 4 major categories based on the location of the monitoring types. We were surprised to find that loading and seepage were categorized as external monitoring when they should have been categorized as internal monitoring.

This work provides insights into the effects of correlation and clustering between monitoring types on reservoir safety, and serves as a reference for reservoir safety prediction. This study provides some theoretical guidance for understanding the correlation between water-level changes in the reservoir area and the type of monitoring at different locations. However, the magnitude of the effect of a particular type of monitoring on the safety of the reservoir area is not yet known, and more in-depth studies are needed to explore the main components that affect the safety of the reservoir area.

## Supporting information

**S1 Dataset.   Comprehensive Safety Monitoring Dataset for Reservoir Areas.**
(XLSX)

## Author contributions

**Data curation:** Weixing Yang, Tingting Li, Bo Wen, Yuan Ren.

**Investigation:** Weixing Yang, Bo Wen, Yuan Ren.

**Methodology:** Weixing Yang, Tingting Li.

**Writing – original draft:** Weixing Yang, Tingting Li.

**Writing – review & editing:** Weixing Yang, Tingting Li.

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
