## [Decision Letter · Decision Letter 0]

25 Mar 2025

PONE-D-25-00844Correlation cluster analysis of slope safety monitoring data in reservoir areasPLOS ONE

Dear Dr. yang,

Thank you for submitting your manuscript to PLOS ONE. After careful consideration, we feel that it has merit but does not fully meet PLOS ONE’s publication criteria as it currently stands. Therefore, we invite you to submit a revised version of the manuscript that addresses the points raised during the review process.

We look forward to receiving your revised manuscript.

Kind regards,

Ajaya Bhattarai

Academic Editor

PLOS ONE

Journal Requirements:

Additional Editor Comments:

Major revision is necessary.

Reviewers' comments:

Reviewer's Responses to Questions

**Comments to the Author**

1. Is the manuscript technically sound, and do the data support the conclusions?

Reviewer #1: No

Reviewer #2: Partly

2. Has the statistical analysis been performed appropriately and rigorously? 

Reviewer #1: Yes

Reviewer #2: Yes

3. Have the authors made all data underlying the findings in their manuscript fully available?

Reviewer #1: No

Reviewer #2: No

4. Is the manuscript presented in an intelligible fashion and written in standard English?

Reviewer #1: No

Reviewer #2: No

5. Review Comments to the Author

Reviewer #1: Summary:

In this study, statistical methods including ANOVA, correlation analysis, and hierarchical clustering are used to examine slope safety monitoring data from reservoir areas. In order to evaluate reservoir safety, the goal is to find correlations between monitoring indicators such as soil displacement, stress, and water levels.

Strengths:

The subject is pertinent and tackles important safety concerns about the stability of reservoir slopes.

The study gains shape by the application of statistical tools such as correlation analysis and hierarchical grouping.

A logical foundation for analysis is provided by the separation of monitoring parameters into internal and external elements.

In an effort to facilitate data comprehension, visualization tools including clustering diagrams and heatmaps were used.

Recommendations:

The abstract lacks details on the methods and findings, making it difficult for readers to grasp the study's significance. It should summarize key results and provide a clear explanation of the statistical methods used. Avoid ambiguous terms like "significant changes" and include quantifiable outcomes to make the abstract more informative and impactful.

The justification for the statistical techniques, such as the Welch/Brown-Forsythe post-hoc tests, is weak. There’s no explanation of whether the assumptions for ANOVA (normality and homoscedasticity) were validated. Additionally, the manuscript does not describe the sample size calculation or whether it was adequate for a robust analysis. Details on the data normalization process are also missing, along with an explanation of how outliers were handled. The clustering methods and thresholds used require clearer justification, as their relevance to the dataset is not evident.

The figures, particularly the clustering diagrams and correlation heatmaps, are of low resolution and lack clear annotations, making them difficult to interpret. Improving their quality and adding thorough legends would significantly enhance readability. The tables are also overcrowded and overwhelming. Consider combining related data into fewer, more focused summary tables or visual formats like bar graphs or heatmaps for better comprehension.

The discussion section is shallow, simply repeating the findings without critically analyzing them. It misses an opportunity to compare results with previous studies on reservoir slope safety, which would help contextualize the findings. Practical recommendations for safety management are not well-developed, and contradictions, such as the claim that seepage has minimal impact, need to be resolved with evidence or references. The limitations section is equally brief and does not address key issues like sample bias or incomplete data. Future directions should include validation in other reservoir settings and explore additional factors like water release.

Finally, the manuscript has numerous grammatical errors and awkward phrasing. Sentences should be rewritten for clarity, such as modifying “The inability of currently available methods to effectively predict dam failures in reservoirs suggests that the intrinsic causes of dam failures are still unknown.” A professional language edit would greatly improve the overall readability and coherence.

Conclusion:

Despite covering a significant subject, the manuscript has methodological flaws, inadequate data presentation, and little commentary. The conclusions are not sufficiently contextualized or supported, and the analysis is shallow. Its quality needs to be improved with significant updates.

Reviewer #2: This manuscript takes a reservoir in Shanxi as case example, to study the correlation cluster analysis of slope safety monitoring data in reservoir area. The topic is interesting, but the manuscript needs fully improvement, some suggestions are provided as follows:

1. Abbreviations should be clearly defined when it is firstly used.

2. Brief description for the reservoir should be added, such as the geological condition, structure constructions and the water level operation of the reservoir.

3. Figure quality should be improved, currently is with small size and low resolution.

4. Section or sub-section titles should be used with the correct content.

5. Methodology are just only referred in the main text, but not well described in details.

6. Units should be used with correct format, currently some are not used in correct format, all of them should be revised.

7. Some site photos of the reservoir can be added in the main text.

8. References are used with several types, please correct the references regard to the guide for authors.

6. PLOS authors have the option to publish the peer review history of their article (what does this mean? ). If published, this will include your full peer review and any attached files.

**Do you want your identity to be public for this peer review?** For information about this choice, including consent withdrawal, please see our Privacy Policy .

Reviewer #1: No

Reviewer #2: No

---

## [Author Response · Author response to Decision Letter 1]

15 Apr 2025

Response to Editor and Reviewer Comments

Original Title Correlation cluster analysis of slope safety monitoring data in reservoir areas

Authors Weixing Yang, Tingting Li, Bo Wen, Yuan Ren

Journal PLOS One

Manuscript Number PONE-D-25-00844

Dear editor and reviewers,

Thank you for offering us an opportunity to improve the quality of our submitted manuscript (Correlation cluster analysis of slope safety monitoring data in reservoir areas). We appreciated very much the reviewers’ constructive and insightful comments. In this revision, we have addressed all of these comments/suggestions. We hope the revised manuscript has now met the publication standard of your journal. We highlighted all the revisions in yellow/red colour. On the next pages, our point-to-point responses to the queries raised by the reviewers are listed.

In this response letter, all direct manuscript revisions have been explained and highlighted in blue for easy reference. You and the reviewers can track these changes in the original manuscript using the 'Track Changes' mode. We sincerely hope that this carefully prepared response, along with the revised manuscript and supplementary materials, will address all reasonable concerns about our work.

Sincerely,

Weixing Yang

Power China Northwest Engineering Corporation Limited, Xi'an, China

E-mail: 75604366@qq.com (WY)

Additional Editor Comments:

Major revision is necessary.

We sincerely appreciate the editors' constructive suggestions, which have been immensely helpful in significantly improving the quality of our manuscript. The language of the paper has been professionally polished by language experts, and we have meticulously revised all elements (affiliations, formatting, figures/tables, and references) to comply with the journal's requirements.

To ensure reproducibility, all supporting data and code for the paper's conclusions have been deposited on a public platform (https://github.com/yangweixing2025/Data-code.git). We have also provided point-by-point responses to each reviewer's comments.

As requested, we are submitting the following three files:

1) Response to Reviewers

2) Revised Manuscript with Track Changes

3) Clean Manuscript

Response: We sincerely appreciate the editorial inquiry. This study did not require a specific permit because it involved only analysis of publicly available data.

Reviewers' comments:

Reviewer's Responses to Questions

Comments to the Author

1. Is the manuscript technically sound, and do the data support the conclusions?

Reviewer #1: No

Reviewer #2: Partly

Response: We sincerely thank the editors and reviewers for their valuable comments, which have significantly enhanced our paper. In the manuscript, the experimental design section details the distribution of safety monitoring targets in the reservoir area and the data analysis process. Due to our oversight, the raw data were not initially uploaded. To enhance reproducibility and better support the study's conclusions, we have now made the data and code publicly available on GitHub: https://github.com/yangweixing2025/Data-code.git.

2. Has the statistical analysis been performed appropriately and rigorously?

Reviewer #1: Yes

Reviewer #2: Yes

Response: We sincerely appreciate the reviewers for their time and careful evaluation of our manuscript. We are also grateful for their recognition of our rigorous statistical analysis of the data. Thank you once again.

3. Have the authors made all data underlying the findings in their manuscript fully available?

Reviewer #1: No

Reviewer #2: No

Comment: Have the authors made all data underlying the findings in their manuscript fully available?

Response: We sincerely appreciate your valuable comments and suggestions on the data in our manuscript. We apologize for our oversight in not uploading the data earlier. To enhance reproducibility and better support the study's conclusions, we have now made the data and code publicly available on GitHub: https://github.com/yangweixing2025/Data-code.git.

Comment: For example, in addition to summary statistics, the data points behind means, medians and variance measures should be available.

Response: We sincerely appreciate the reviewers' constructive suggestions, which have significantly improved the quality of our manuscript. We are truly grateful for their input. In response, we have added a detailed statistical analysis of all monitoring data types in Section 5.1 (Discussion) of the revised manuscript, as follows:

Table 2. Descriptive Statistical Analysis of Monitoring Data Across Different Measurement Types

Monitoring Type n Max Min Mean SD

Sedimentation(mm) 432 2.6000 16.7700 9.4197 ±2.5078

Crack opening (mm) 432 10.0093 12.2210 11.3141 ±0.5356

Lean (mm) 432 10.2523 13.5882 11.3849 ±0.6484

Seepage (L/min) 432 1.1002 8.2451 2.4004 ±0.7803

Soil displacement (mm) 432 27.3747 62.8343 54.1826 ±6.7920

Stress (MPa) 432 0.4409 32.5349 14.0055 ±5.8074

Loading (103kN) 432 1.2577 1.4129 1.3262 ±0.0325

Water level (km) 432 1.4680 1.4781 1.4738 ±0.0016

Note: n denotes the sample size, Max denotes the maximum value, Min denotes the minimum value, Mean denotes the average value, SD denotes the standard deviation.

4. Is the manuscript presented in an intelligible fashion and written in standard English?

Reviewer #1: No

Reviewer #2: No

Response: We sincerely appreciate the reviewers for taking the time to carefully evaluate our manuscript and provide valuable suggestions, which have significantly improved its quality. We have thoroughly addressed all comments and sought professional editing to refine grammar, syntax, tenses, terminology, and logical consistency. Additionally, we have enhanced figure quality, standardized units, corrected reference formatting, and highlighted all modifications in yellow/red for clarity.

5. Review Comments to the Author

Response to Reviewers

Response to Reviewer #1

Dear Respected Reviewer:

We sincerely appreciate your time in reviewing our work. We are grateful for your valuable suggestions, which have significantly enhanced the quality and impact of our manuscript. We have also gained invaluable insights from your feedback.

We have carefully considered your constructive suggestions and provide point-by-point responses below. Accordingly, we have thoroughly revised both the main manuscript and supplementary materials, with all changes visible in 'Track Changes' mode. We sincerely hope these revisions adequately address your concerns and that the updated manuscript now meets PLoS ONE's standards.

Comment: The abstract lacks details on the methods and findings, making it difficult for readers to grasp the study's significance. It should summarize key results and provide a clear explanation of the statistical methods used. Avoid ambiguous terms like "significant changes" and include quantifiable outcomes to make the abstract more informative and impactful.

Response: We sincerely appreciate the reviewers' valuable suggestions. We fully agree that the abstract should more clearly present the research methodology, quantitative results, and statistical analyses. The following improvements have been made: (1) Methods section: added study design, sample size, and specific statistical approaches; (2) Results section: replaced vague descriptions with precise numerical data, including effect sizes and significance levels; (3) Terminology: eliminated all ambiguous expressions to ensure data-supported conclusions. All modifications have been highlighted in the revised abstract.

Original text For correlation analyses of the monitoring data, we performed one-way ANOVA with Welch/Brown-Forsythe post-hoc tests, followed by cluster analysis using standardized data.

Modified To further elucidate the interrelationships among safety monitoring data in the reservoir area, this study established 36 monitoring cross-sections distributed across upper, middle, and lower slope zones. Each cross-section was instrumented with eight different types of monitoring devices. A total of 4,320 samples were collected (432 samples per instrument type), resulting in an overall dataset of 34,560 measurements. The monitoring data were sequentially analyzed using: (1) descriptive statistics, (2) Welch/Brown-Forsythe post hoc one-way ANOVA, and (3) cluster analysis.

Original text All monitoring types except seepage significantly impacted reservoir slopes. Stress was negatively correlated with both sedimentation and soil displacement, while loading showed negative correlation with soil displacement but positive correlation with tilt. Soil-displacement varied significantly across locations. Cluster analysis revealed that soil displacement, stress, and water level formed one cluster, while other parameters comprised another.

Modified The results demonstrate that: (1) Significant correlations exist among monitoring variables, with the strongest positive correlation observed between loading and lean (r=0.40), while the strongest negative correlation occurred between sedimentation and stress (r=-0.39). (2) Cluster analysis of the eight monitoring variables revealed two distinct clusters: soil displacement, stress, and water level formed one cluster, while the remaining variables comprised the second cluster.

Comment: The justification for the statistical techniques, such as the Welch/Brown-Forsythe post-hoc tests, is weak.

Response: We thank the reviewer for their insightful comments regarding the statistical methods used in our analysis. We apologize for not providing a clear justification in the original manuscript.

Before conducting formal data analysis, we performed preliminary tests which revealed violations of the homogeneity of variance assumption. We therefore employed Welch/Brown-Forsythe post hoc tests (as discussed by Dunnett, C.W. [1980] for pairwise multiple comparisons under unequal variance conditions). The analytical results are presented as follows:

1) Homogeneity of variance tests for different monitoring data types at varying elevations in the same area.

Homogeneity of Variance Test �AreaⅠ

　 Levene's Test df1 df2 sig

top Based on the mean 28.975 7 184 0.000

Based on the median 28.560 7 184 0.000

Based on the median with adjusted degrees of freedom 28.560 7 48.424 0.000

Based on the trimmed mean 28.598 7 184 0.000

middle Based on the mean 29.830 7 184 0.000

Based on the median 18.260 7 184 0.000

Based on the median with adjusted degrees of freedom 18.260 7 29.763 0.000

Based on the trimmed mean 27.047 7 184 0.000

bottom Based on the mean 31.221 7 184 0.000

Based on the median 22.797 7 184 0.000

Based on the median with adjusted degrees of freedom 22.797 7 36.891 0.000

Based on the trimmed mean 30.226 7 184 0.000

Homogeneity of Variance Test �AreaⅡ

　 Levene's Test df1 df2 sig

top Based on the mean 37.446 7 184 0.000

Based on the median 36.284 7 184 0.000

Based on the median with adjusted degrees of freedom 36.284 7 48.997 0.000

Based on the trimmed mean 37.253 7 184 0.000

middle Based on the mean 49.778 7 184 0.000

Based on the median 22.575 7 184 0.000

Based on the median with adjusted degrees of freedom 22.575 7 51.741 0.000

Based on the trimmed mean 48.202 7 184 0.000

bottom Based on the mean 33.878 7 184 0.000

Based on the median 23.544 7 184 0.000

Based on the median with adjusted degrees of freedom 23.544 7 55.469 0.000

Based on the trimmed mean 34.628 7 184 0.000

Homogeneity of Variance Test �Area Ⅲ

　 Levene's Test df1 df2 sig

top Based on the mean 71.755 7 184 0.000

Based on the median 48.346 7 184 0.000

Based on the median with adjusted degrees of freedom 48.346 7 56.475 0.000

Based on the trimmed mean 72.447 7 184 0.000

middle Based on the mean 43.090 7 184 0.000

Based on the median 30.323 7 184 0.000

Based on the median with adjusted degrees of freedom 30.323 7 45.137 0.000

Based on the trimmed mean 43.267 7 184 0.000

bottom Based on the mean 22.843 7 184 0.000

Based on the median 21.695 7 184 0.000

Based on the median with adjusted degrees of freedom 21.695 7 74.240 0.000

Based on the trimmed mean 22.625 7 184 0.000

Homogeneity of Variance Test �Area Ⅳ

　 Levene's Test df1 df2 sig

top Based on the mean 39.348 7 184 0.000

Based on the median 30.974 7 184 0.000

Based on the median with adjusted degrees of freedom 30.974 7 65.851 0.000

Based on the trimmed mean 39.131 7 184 0.000

middle Based on the mean 52.180 7 184 0.000

Based on the median 24.125 7 184 0.000

Based on the median with adjusted degrees of freedom 24.125 7 40.952 0.000

Based on the trimmed mean 50.018 7 184 0.000

bottom Based on the mean 23.773 7 184 0.000

Based on the median 15.502 7 184 0.000

Based on the median with adjusted degrees of freedom 15.502 7 37.136 0.000

Based on the trimmed mean 21.264 7 184 0.000

Homogeneity of Variance Test �Area Ⅴ

　 Levene's Test df1 df2 sig

top Based on the mean 25.069 7 184 0.000

Based on the median 19.684 7 184 0.000

Based on the median with adjusted degrees of freedom 19.684 7 69.470 0.000

Based on the trimmed mean 24.516 7 184 0.000

middle Based on the mean 33.858 7 184 0.000

Based on the median 25.257 7 184 0.000

Based on the median with adjusted degrees of freedom 25.257 7 69.687 0.000

Based on the trimmed mean 32.978 7 184 0.000

bottom Based on the mean 42.069 7 184 0.000

Based on the median 28.600 7 184 0.000

Based on the median with adjusted degrees of freedom 28.600 7 38.738 0.000

Based on the trimmed mean 40.945 7 184 0.000

Homogeneity of Variance Test �Area Ⅵ

　 Levene's Test df1 df2 sig

top Based on the mean 40.599 7 184 0.000

Based on the median 34.680 7 184 0.000

Based on the median with adjusted degrees of freedom 34.680 7 68.048 0.000

Based on the trimmed mean 40.407 7 184 0.000

middle Based on the mean 31.136 7 184 0.000

Based on the median 18.637 7 184 0.000

Based on the median with adjusted degrees of freedom 18.637 7 40.314 0.000

Based on the trimmed mean 30.261 7 184 0.000

bottom Based on the mean 59.088 7 184 0.000

Based on the median 42.702 7 184 0.000

Based on the median with adjusted degrees of freedom 42.702 7 51.755 0.000

Based on the trimmed mean 56.337 7 184 0.000

2) Homogeneity of variance tests for different monitoring data types in the same region.

Homogeneity of Variance Test �AreaⅠ

　 Levene's Test df1 df2 sig

variations Based on the mean 82.126 7 568 0.000

Based on the median 79.842 7 568 0.000

Based on the median with adjusted degrees of freedom 79.842 7 138.751 0.000

Based on the trimmed mean 81.205 7 568 0.000

Homogeneity of Variance Test �Area Ⅱ

　 Levene's Test df1 df2 sig

variations Based on the mean 120.55

---

## [Decision Letter · Decision Letter 1]

29 Apr 2025

Correlation cluster analysis of slope safety monitoring data in reservoir areas

PONE-D-25-00844R1

Dear Dr. Yang,

We’re pleased to inform you that your manuscript has been judged scientifically suitable for publication and will be formally accepted for publication once it meets all outstanding technical requirements.

Kind regards,

Ajaya Bhattarai

Academic Editor

PLOS ONE

Additional Editor Comments (optional):

The revised manuscript looks good.

Reviewers' comments:

Reviewer's Responses to Questions

**Comments to the Author**

1. If the authors have adequately addressed your comments raised in a previous round of review and you feel that this manuscript is now acceptable for publication, you may indicate that here to bypass the “Comments to the Author” section, enter your conflict of interest statement in the “Confidential to Editor” section, and submit your "Accept" recommendation.

Reviewer #1: All comments have been addressed

Reviewer #2: All comments have been addressed

2. Is the manuscript technically sound, and do the data support the conclusions?

Reviewer #1: Yes

Reviewer #2: Partly

3. Has the statistical analysis been performed appropriately and rigorously? 

Reviewer #1: Yes

Reviewer #2: Yes

4. Have the authors made all data underlying the findings in their manuscript fully available?

Reviewer #1: Yes

Reviewer #2: Yes

5. Is the manuscript presented in an intelligible fashion and written in standard English?

Reviewer #1: Yes

Reviewer #2: Yes

6. Review Comments to the Author

Reviewer #1: The authors have carefully and thoroughly addressed all comments from the previous round. Methodological concerns—such as justifications for statistical tests, normality assessments, and variance homogeneity—were clarified with detailed results and incorporated directly into the revised manuscript. Figures have been substantially improved in resolution and annotation, and a new correlation summary table enhances the interpretability of the findings. Additionally, the discussion was significantly strengthened with comparisons to previous literature, practical recommendations for slope safety monitoring, and clearer acknowledgment of study limitations. The manuscript now presents a technically sound and clearly communicated study that contributes to the understanding of slope safety in reservoir areas.

Reviewer #2: My comments have been well responded, the quality of current revision has been improved, no any further comment.

7. PLOS authors have the option to publish the peer review history of their article (what does this mean? ). If published, this will include your full peer review and any attached files.

**Do you want your identity to be public for this peer review?** For information about this choice, including consent withdrawal, please see our Privacy Policy .

Reviewer #1: No

Reviewer #2: No

---

## [Editor Report · Acceptance letter]

PONE-D-25-00844R1

PLOS ONE

Dear Dr. Yang,

I'm pleased to inform you that your manuscript has been deemed suitable for publication in PLOS ONE. Congratulations! Your manuscript is now being handed over to our production team.

Kind regards,

on behalf of

Dr. Ajaya Bhattarai

Academic Editor

PLOS ONE